# Short communication: Analytical models for 2D landscape evolution

Philippe Steer[1]

[1]Univ Rennes, CNRS, Géosciences Rennes - UMR 6118, F-35000 Rennes, France.

*Correspondence to*: Philippe Steer (philippe.steer@univ-rennes1.fr)

**Abstract.** Numerical modelling offers a unique approach to understand how tectonics, climate and surface processes govern landscape dynamics. However, the efficiency and accuracy of current landscape evolution models remain a certain limitation. Here, I develop a new modelling strategy that relies on the use of 1D analytical solutions to the linear stream power equation to compute the dynamics of landscapes in 2D. This strategy uses the 1D ordering, by a directed acyclic graph, of model nodes based on their location along the water flow path to propagate topographic changes in 2D. I demonstrate that this analytical model can be used to compute in a single time step, with an iterative procedure, the steady-state topography of landscapes subjected to river, colluvial and hillslope erosion. This model can also be adapted to compute the dynamic evolution of landscapes under either heterogeneous or time-variable uplift rate. This new model leads to slope-area relationships exactly consistent with predictions and to the exact preservation of knickpoint shape throughout their migration. Moreover, the absence of numerical diffusion or of an upper bound for the time step offer significant advantages compared to numerical models. The main drawback of this novel approach is that it does not guarantee the time-continuity of the topography through successive time steps, despite practically having little impact on model behaviour.

## 1 Introduction

While the elevated but incised landscapes of mountain belts testify to the cumulated actions of tectonics, erosion and climate, unravelling how these processes act and interact to shape the Earth's surface remains one of the most challenging issues in Earth Sciences (e.g. Molnar & England, 1990; Willett, 1999; Whipple, 2009; Steer et al., 2014; Croissant et al., 2019). Numerical models have been pivotal to understanding how topography and erosion respond to spatial and temporal changes in climate and tectonics (e.g. Howard et al., 1994; Whipple & Tucker, 1999; Tucker & Whipple, 2002; Carretier & Lucazeau, 2005; Thieulot et al., 2014; Croissant et al., 2017). At the mountain-scale, numerical models generally account for geomorphological processes using effective and reduced-complexity erosion laws such as the stream power incision model (SPIM) for rivers (e.g. Howard et al., 1994) and diffusion for hillslopes (e.g. Roering et al., 1999). In particular, the SPIM is popular in landscape evolution models (LEM) as its physical expression resolves to a non-linear kinematic wave equation, which offers simple finite difference or finite volume solutions in 1 and 2 D (e.g. Pelletier, 2008; Braun & Willett, 2013; Campforts & Govers, 2015; Campforts et al., 2017). Despite these benefits, these numerical solutions have several drawbacks:

1) their stability or consistency requires the use of a small time step that must respect the Courant condition, i.e. that an erosional wave cannot travel over a distance greater than one or a few node spacing during one time step; 2) and they are prone to numerical diffusion and therefore only offer approximate solutions. 2D Numerical schemes have recently been developed to reduce the time-step dependency to grid spacing (Braun & Willett, 2013) or numerical diffusion (Campforts & Govers, 2015). In 1D, evolution of river profiles can be derived using analytical solutions determined by the method of the characteristics (Luke, 1972, 1974, 1976; Weissel & Seidl, 1998; Whipple & Tucker, 1999; Lavé, 2005; Pritchard, 2009; Royden & Perron 2013). These solutions have been successfully used in formal inversion of river profiles (Goren et al., 2014a; Fox et al., 2014; Goren, 2016), but have been largely ignored in forward landscape evolution models, despite their inherent exact accuracy. This likely results from the apparent absence of analytical solution in 2D.

In this study, I extend the applicability of these 1D analytical solutions to 2D problems by developing a new type of landscape evolution model based on analytical solutions. I first demonstrate how this model, that I refer to as Salève, can be used to compute – in a single time step - a steady-state topography in 2D. I then develop a dynamic version of Salève to solve for transient landscape changes under heterogeneous or time-variable uplift. Last, I demonstrate the ability of Salève to accurately model the propagation of knickpoints in LEMs and to account for river, colluvial and hillslope erosion.

## 2 From a 1D to a 2D analytical solution to the stream power law

Most LEMs require the computation of river water discharge as the main driver of river erosion and sediment transport. While physical-based flow algorithms offer more accurate solutions (e.g. Davy et al., 2017), water routing in 2D LEMs is generally achieved using simple flow algorithms, like the steepest slope (O'Callaghan & Mark, 1984) or the multi-flow direction (Quinn et al., 1991; Freeman, 1991). The Fastscape algorithm, and other graph-based approaches, offer a very efficient mean to order nodes along the steepest water flow path and to compute river discharge and drainage area (Braun & Willett, 1993; Schwanghart & Scherler, 2014). A single receiver and potentially several donors are attributed to each node of the topographic grid, to recursively build a node stack (or graph) from the outlet node to the crest nodes of each catchment. Each node is therefore associated to its outlet node through a single flow path. These flow paths represent 2D trajectories in the $(x, y)$ space, that can be converted to pseudo 1D trajectories (i.e. to directed acyclic graphs) using the node ordering of the stack. For instance, local river slope along the water flow can be computed by simply differentiating elevation over the distance along the river $l$ between successive nodes. 2D LEMs solving for river erosion using a single flow algorithm and local river slope or water discharge are therefore fundamentally solving a 1D problem, based on a 2 D description of the flow. To be more accurate, they actually solve for a series of 1D problems, with one 1D problem for each catchment connected to an outlet.

In 1D, a classical detachment-limited approach to describe the rate of change in river elevation change $z$ with time $t$ is the SPIM (Howard and Kerby, 1983; Howard, 1994; Whipple & Tucker, 1999; Lague, 2014):

$$\frac{\partial z(l,t)}{\partial t} = U(l,t) - K'(l)Q_w(l)^m \left(\frac{\partial z(l,t)}{\partial l}\right)^n, \tag{1}$$

where $U$ is the uplift rate, $K'$ the erodability, $Q_w = rA$ the water discharge with $A$ the drainage area, $r$ the mean daily runoff, and $m$ and $n$ are two exponents. This equation can be cast in a more commonly used form, as a function of drainage area, by defining an effective erodability $K = K'r^m$:

$$\frac{\partial z(l,t)}{\partial t} = U(l,t) - K(l)A(l)^m \left(\frac{\partial z(l,t)}{\partial l}\right)^n, \tag{2}$$

5    This equation corresponds to a non-linear kinematic wave equation with a celerity $C(l) = K(l)A(l)^m(\partial z(l,t)/\partial l)^{n-1}$ representing the speed at which information propagates along the river (e.g. Rosenbloom & Anderson, 1994; Weissel & Seidl, 1998; Whipple & Tucker, 1999; Royden & Perron, 2013). Following Royden & Perron (2013), this migrating information can be referred to as slope patches. Integrating the invert of this celerity along the river path, from the river outlet at $l = 0$ to a point of coordinate $l$ along the river, defines the river response time:

$$\tau(l) = \int_0^l \frac{1}{C(l')} dl' = \int_0^l \frac{1}{K(l')A(l')^m(\partial z(l',t)/\partial l')^{n-1}} dl', \tag{3}$$

Using this response time and assuming a constant but potentially heterogeneous uplift rate $U(l)$ or a uniform but potentially variable uplift rate $U(t)$, river profile elevation can be derived analytically assuming A is known (see derivation in Royden & Perron, 2013). As I intend to implement a solution in a LEM, the solution needs to remain practical. In particular, it is noticeable that the response time and celerity become independent of local river slope $S(l) = \partial z(l,t)/\partial l$ when $n = 1$, which is a classical

15 choice in forward or invert landscape evolution models (e.g. Goren et al., 2014a; Fox et al., 2014). Under this condition, and assuming a constant and homogeneous uplift rate $U$, the steady-state river profile elevation is:

$$z(l) = z(0) + U\tau(l) = z(0) + U \int_0^l \frac{1}{KA(l')^m} dl', \text{with } z(0) = z_{base}, \tag{4}$$

with $z_{base}$ the base-level elevation. Note that this solution is asynchronous as steady-state is achieved for an increasing response time in the upstream direction. Importantly, as the flow network is not known a priori, this integral solution still

20 requires to numerically compute a flow network and drainage area over a discretized grid. In the following, I adapt this formalism to develop two modelling approaches which computes either the steady-state topography of a landscape or solves for its dynamic evolution (Fig. 1).

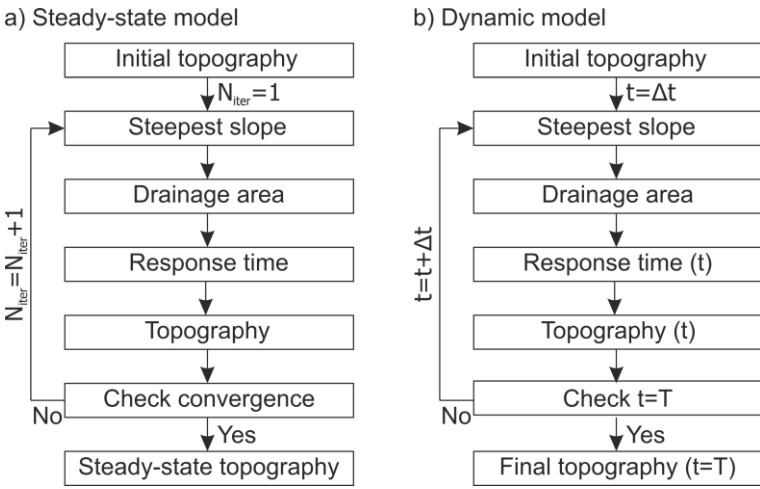

**Figure 1.** Overview of the algorithms used for the a) steady-state and b) dynamic simulations.

## 3 A single time step iterative solution to topographic steady-state in 2D

This solution (Eq. 4) can be extended to spatially variable uplift rate $U(l)$ by simply using the response time of the receiver node $\tau_R(l)$ and its elevation $z_R(l)$ :

$$z(l) = z_R(l) + U(l)\,(\tau(l) - \tau_R(l))\text{ for }l > 0\text{, and }z(0) = z_{base}, \tag{5}$$

Obviously, this operation needs to be performed iteratively and in the correct node order, from the outlet node towards the upstream direction using the node stack or graph (Braun & Willett, 2013; Schwanghart & Scherler, 2014). Ignoring hillslope processes, I use this solution to attempt computing with Salève in a single iteration the steady-state topography (Fig. 2). The initial topography consists in a flat surface with a random noise discretized by a regular grid. I use $m = 0.5$, corresponding to the classical unit stream power, $U = 10\text{ mm.yr}^{-1}$, $K' = 1\ 10^{-6}\text{ yr}^{-1}$, $r = 5/365\text{ m.day}^{-1}$ and a square model domain of extent $L = 10$ km with a resolution of 50 m, corresponding to $n_{pt} = 40.401$ points. Flow over the topography is computed using the single-flow algorithm provided by Topotoolbox (Schwanghart & Scherler, 2014), which efficiently exploits the directed acyclic graph structure of the flow network (Phillips et al., 2015).

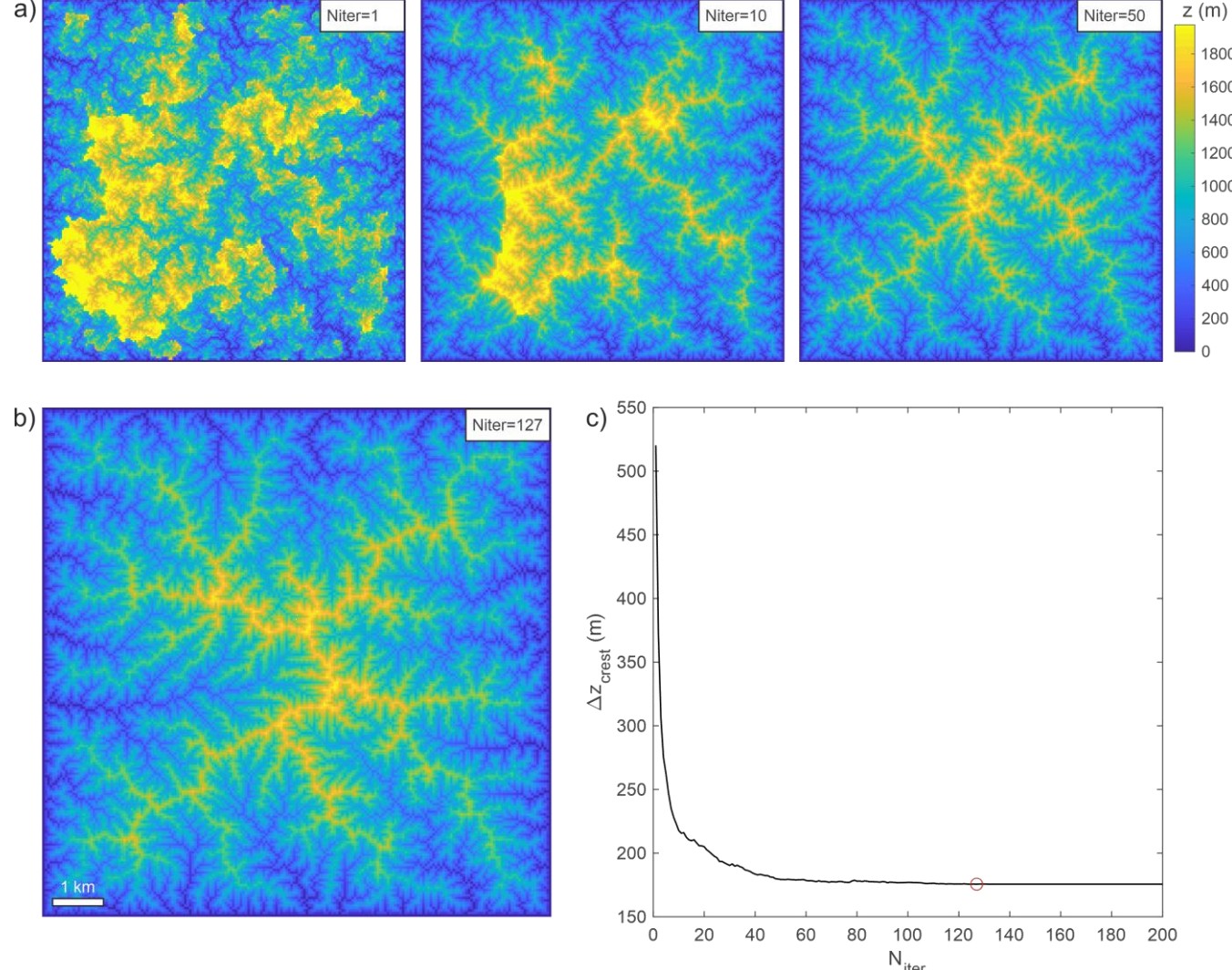

**Figure 2.** Modeled steady-state topographies obtained after a) 1 (left), 10 (middle) and 50 (right) iterations. b) The steady-state topography is obtained after 127 iterations. c) Convergence of the iterative algorithm inferred from the degree of crest disequilibrium $\Delta z_{crest}$, computed as the average of the absolute difference of elevation between crest nodes of juxtaposing catchments. Red dot indicates model shown in panel b. Note that in panel a, the colormap is bounded by the maximum elevation of the steady-state topography shown in panel b.

The obtained solution looks very roughly like a classical steady-state topography, and yet is not strictly at steady-state (Fig. 1a). Indeed, during this first iteration, the used scheme (Fig. 1a) imposes that rivers develop over the flow network defined by the initial topography and, in turn, does not ensure that the nodes located on the same crest of two juxtaposing catchments share the same response time or the same elevation. This leads to an excessive elevation as some rivers have planar length greater than predicted. This is the main limit of this 1D algorithm that cannot ensure the optimality of the 2D organization of the river network at steady-state after only one iteration (Fig. 2).

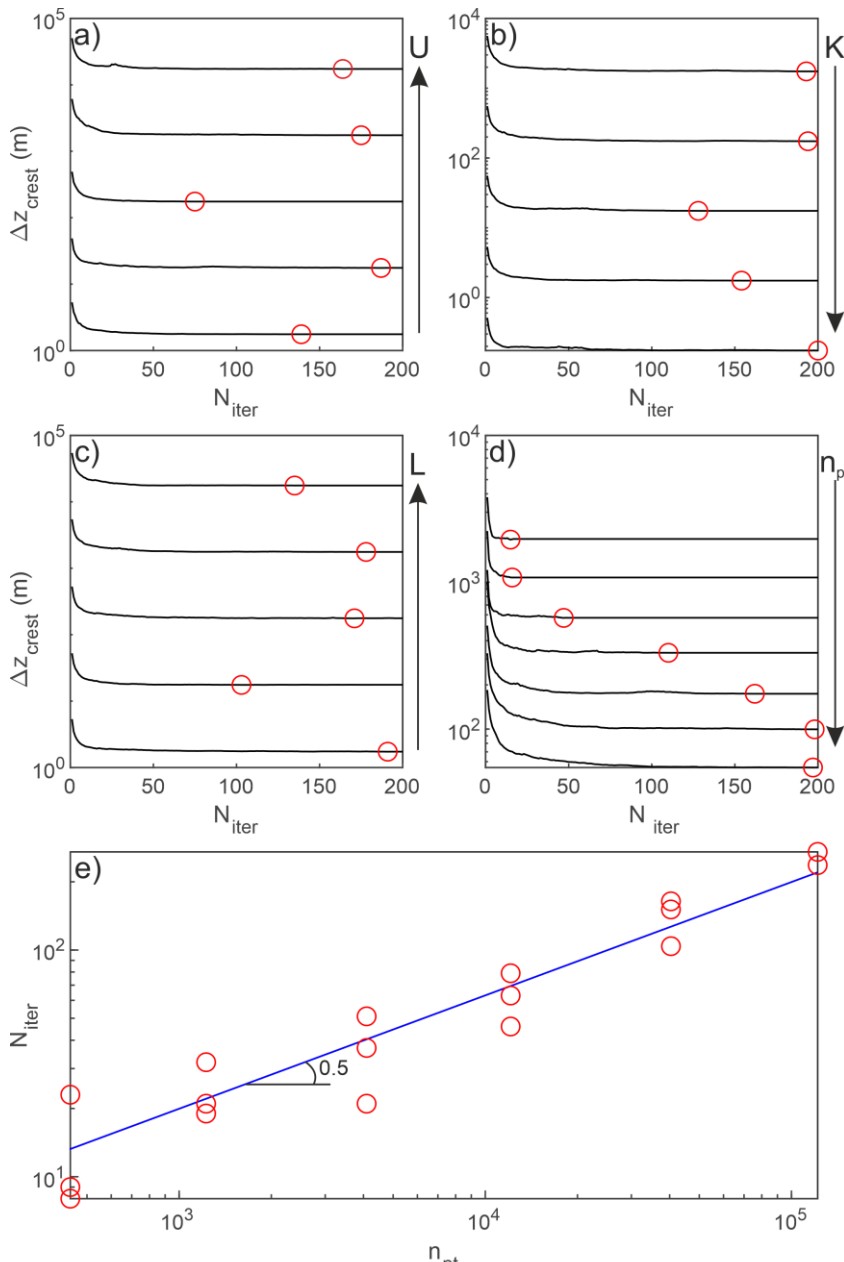

**Figure 3.** Influence of model parameters and geometry on the convergence towards a steady-state landscape. a) Uplift rate $U$ was varied between $10^{-4}$ and 1 m.yr$^{-1}$. b) Erodability $K$ was varied between $1\ 10^{-7}$ and $1\ 10^{-3}$ yr$^{-1}$. c) Model length $L$ was varied between 0.1 and 100 km. d) The number of model points $n_{pt}$ was varied between $1.2\ 10^3$ and $0.4\ 10^6$. e) The relationship between the number of iterations required to reach steady-state and $n_{pt}$ follows a power law with an exponent 0.5 (blue line).

However, repeating this operation by computing the topography and then updating the flow network (i.e., by computing the steepest slope, node order and drainage area or discharge) after each iteration leads to a steady-state topography after few tens of iterations $N_{iter}$ (Fig. 2). To assess the convergence of this iterative procedure, I define the degree of crest disequilibrium

$\Delta z_{crest}$ as being the average of the absolute difference of elevation between crest nodes of juxtaposing catchments. I find that $\Delta z_{crest}$ follows a rapid decay with $N_{iter}$ until reaching a slower decay phase when $N_{iter} \geq 40$. $\Delta z_{crest}$ never reaches 0 m, even after 100 iterations, as differences of elevation can remain along the two sides of the crests, as in other LEMs, due to the non-continuity of the spatial discretization for grid-based models (Fig. 2c). However, the model reaches a stable solution at $N_{iter} =$ 127. Note that running the same model but with a different initial topography leads to a variability of this required number of iterations due to the initial configuration of the flow network.

Changing $U$, $K$ and $L$, while keeping $n_{pt}$ constant, does not lead to a significant change in the number of iterations required to reach steady-state (Fig. 3). This shows that the convergence of this algorithm is independent of the model parametrization. However, increasing the number of points $n_{pt}$ lead to an increase in the number of iterations required to reach steady-state which scales with $n_{pt}^{0.5}$, or in other words with the number of nodes in one of the horizontal dimensions (Fig. 3). This scaling emerges due to the more numerous numbers of local (i.e. among direct neighbors) permutations of the crest location required to reach a stable fluvial organization when increasing $n_{pt}$.

The new algorithm developed in Salève presents significant advantages compared to finite difference schemes, which are fundamentally limited by the time step $\Delta t$ that must respect the Courant conditions $\Delta t < k\,\Delta x / \max(C(l))$, with k equals to ~0.1 or to 100 for explicit or implicit schemes, respectively (e.g. Braun & Willett, 2013). Therefore, these finite difference solutions are doomed to use shorter time steps and a larger number of iterations when considering finer resolutions. At the contrary, this analytical LEM converges towards steady-state with roughly the same number of iterations, independently of the celerity $C(l)$, which is set by $K$, A (i.e. $L^2$) and $m$. The number of required iterations however increases $n_{pt}^{0.5}$, which is equivalent to an increase with $\Delta x = L/n_{pt}^{0.5}$ when $L$ is constant, as in classical finite difference schemes. Moreover, this steady-state modelling approach is compatible with spatially variable $U$, $K$ and $r$.

## 4 A 2D dynamical model with analytical accuracy

I now explore the use of this analytical model in dynamic simulations with Salève (Fig. 1b). I first consider the case of potentially heterogeneous but constant uplift rate $U(l)$. A transient solution for river elevation $z(l,t)$ at a specific time $t$ can be computed using equation (4) or (5) by simply thresholding the response time so that for every node $\tau(l,t) = \min(\tau(l), t)$. It results that:

$$z(l,t) = z_R(l,t) + U(l)\,(\tau(l,t) - \tau_R(l,t)) \text{ for } l > 0, \text{ and } z(0,t) = z_{base}. \tag{6}$$

This solution therefore enables to compute the time evolution of a landscape under potentially heterogeneous erodibility, uplift and runoff (or precipitation) rates. Thresholding the response time enforces that the uplift rate is considered null before the beginning of the simulation. The limitation of non-optimality of the planar organization of flow network remains as in the steady-state solution. However, this limitation can be solved by simply updating the river network, the node order, the steepest slope and water discharge after each time step, as in any other LEMs. As the time-step is not constrained by numerical stability

issues, such as the Courant condition, it can be chosen only based on the rate of flow network reorganization linked to river capture and piracy. Note however that in the dynamic Salève models, flow network reorganization will lead to an immediate topographic reorganization, so that to respect Eq. (6). Indeed, time evolution of the elevation in Salève should not be seen as a continuous time evolution of a same topography, that would evolve by erosion under different time and space distribution of water discharge (e.g. as in other LEM), but as a succession of topographic realizations which respect that the distribution of elevation is set by the flow network. In other words, Salève does not fully guarantee the time-continuity of the topography through successive time steps, despite practically having a limited impact on model behavior as I will demonstrate later.

I here run a simulation, using the same parameters as in the steady-state simulation, over a duration of 500 kyr (Fig. 4). The time-step $\Delta t$=2 and 0.2 kyr correspond to about 45 and 4.5 times the Courant condition, respectively. Because implicit finite difference solutions to the SPIM also remain numerically stable for time steps longer than the one imposed by the Courant condition, I also run simulations using an implicit solution with the same parameters and time steps and compare them to the results of the Salève simulations. The implicit solution is computed following Equation 22 in Braun & Willett (2013). I also compare Salève with results obtained with and implicit solution using a $\Delta t$=0.002 kyr, corresponding to a Courant condition of 0.45.

The final topographies, i.e. at steady-state, obtained with Salève or with the implicit solution share roughly the same statistical properties in terms of vertical and horizontal organization. The time evolution of the mean, mean($z$), and maximum, max($z$), elevation is similar in all the models, even if the steady-state value is higher by ~50 for mean($z$) and ~500 m for max($z$), with the implicit solution (Fig. 4c). Moreover, the fluvial network and hence the topography modelled with Salève reach a stable configuration once at steady state, with no subsequent vertical or horizontal changes. Topographic stability occurs when the model time $t$ becomes greater or equal to the response time $\tau(l)$ of all the model nodes, and in particular the ones located on crests (Fig. 4b). This is particularly true if $\Delta t$ is small enough to allow the horizontal organization of the fluvial network to evolve concomitantly with its vertical component. At the contrary, the topography simulated by the finite difference models continue to evolve after steady-state, in particular the maximum elevation max($z$), with larger variations for models with longer time steps, which occur due to catchment re-organization and numerical noise.

Moreover, erosion rates $E$ first increase more slowly and then more rapidly in Salève than with the implicit solutions before reaching steady-state (Fig. 3d). In particular, the second phase is due to longer upstream distances and erosional response times in the topographies simulated with the implicit solution than with Salève (Fig. 3f). This is at least partly due to the dependency of the transient phase duration to $\Delta t$ for finite difference models (Braun & Willett, 2013). Geometrically, longer transient phases are associated to fluvial networks with longer upstream distances, i.e. distances to the outlet, in the implicit models with longer time-steps compared to implicit models with shorter time-steps or to Salève models (Fig. 3e). These results also show that the response time of the landscapes is shorter than with other 2D LEMs as it is equal to the 1D response time, based on the flow network length at steady-state, when the use time step is sufficiently short to allow progressive reorganization of the fluvial network (e.g. model with $\Delta t$=0.2 kyr).

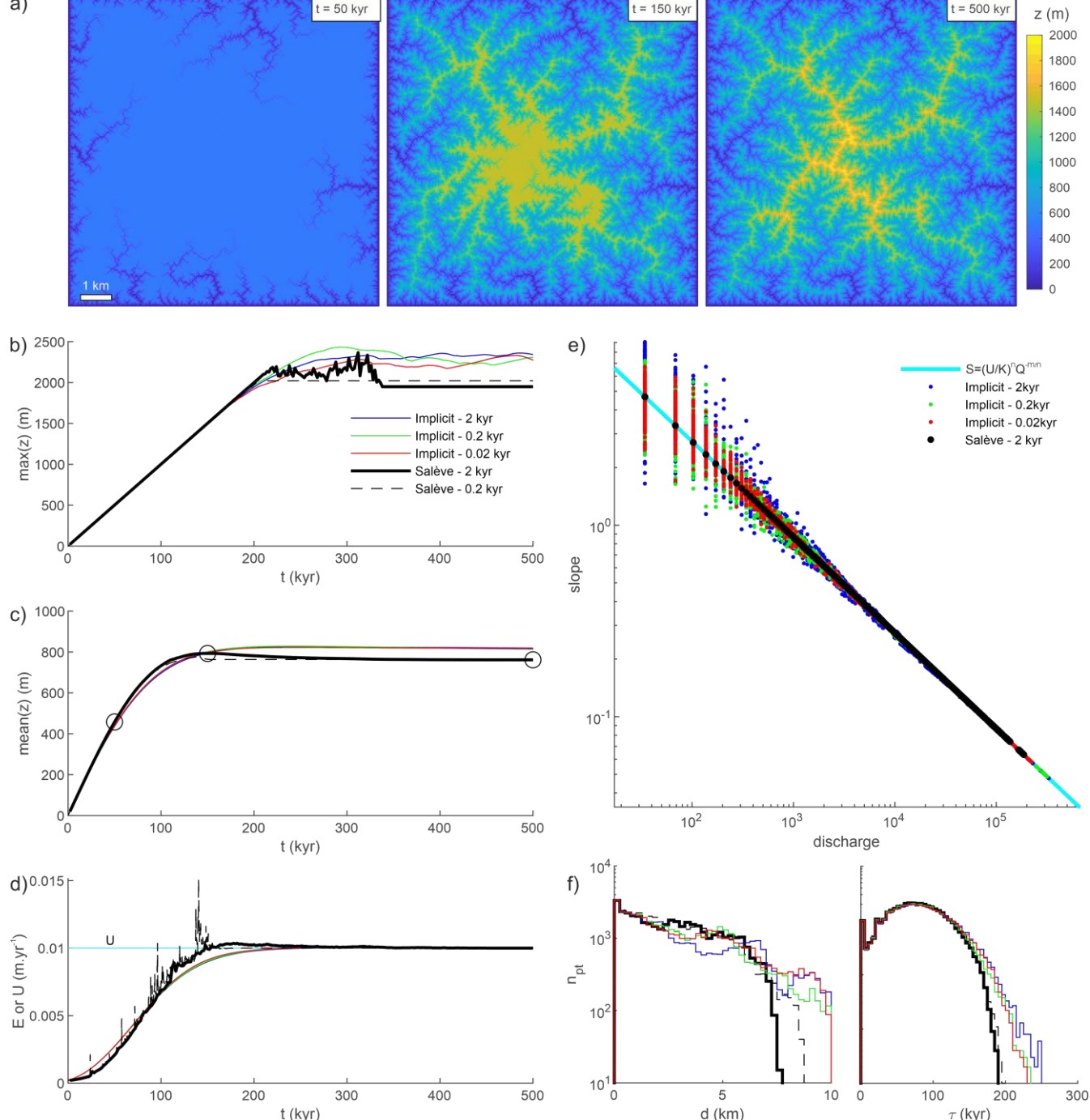

**Figure 4.** Dynamic behavior of the Salève model. a) Time evolution of the modelled topography after 50 (left), 150 (middle) and 500 kyr (right). Time evolution of the b) max elevation, c) mean elevation and d) mean erosion rate for Salève, using a time step of $\Delta t = 2$ kyr (black line) and 0.2 kyr (dashed black line) and for the implicit solution with $\Delta t = 2$ kyr (blue line), 0.2 kyr (green line) and 0.02 kyr (red line). The uplift rate U is shown on panel d with a cyan line. e) Slope-discharge distributions at steady-state (at 500 kyr) for the three models compared

to the predicted relationship (cyan line). No binning is done and all the model nodes are represented on panel d. f) Histogram of upstream distances $\boldsymbol{d}$ (left) and response times $\boldsymbol{\tau}$ (right) for the different models at steady-state.

Erosion rates in Salève, calculated by differentiating elevation between successive time steps and subtracting the contribution of uplift, are significantly more variable, in particular for the model with the shorter time step $\Delta t =0.2$ kyr, than with the implicit solutions. This variability highlights phases of fluvial network reorganization which lead to immediate topographic reorganization, due to time-discontinuity, and therefore to immediate increase in erosion rates.

In terms of horizontal organization, all the Salève and implicit models lead to the same Hack's law (Hack, 1957), which relates through a power law relationship the downstream maximum river length $L_r$ with catchment area $L_r \propto A^h$. A least-square fitting gives an exponent $h$ of 0.65±0.01 for Salève and the implicit models at steady-state, with no dependency over $\Delta t$. In terms of vertical organization, the slope-discharge relationship obtained with Salève at steady-state fits perfectly the predicted one, $S = (U/K)^{1/n} Q^{-m/n}$, while the implicit solution shows a significant spread, in particular at low drainage area or discharge, that increases with $\Delta t$ (Fig. 3d). Using $\Delta t$=0.02 instead of 2 kyr, leads to a slightly better consistency between the implicit and Salève solution, including the slope-discharge relationship and the temporal evolution of elevation.

## 5 Application: Time variable uplift and knickpoint propagation

I now investigate the case of time-variable but homogeneous uplift rate $U(t)$. Following Royden & Perron (2013), this case leads to additional complexity as the uplift rate, when the slope patches were initiated, must be tracked during their upstream migration. In a LEM, this is performed by computing, at the specific time $t$, what I refer to as the uplift memory map $U_{mem}(l,t) = U(t - \tau(l,t))$. It is not equivalent to a classical uplift map and corresponds to the uplift rate when the slope patches were formed. I remind here that the response time is bounded by actual model time $\tau(l,t) = \min(\tau(l),t)$. The elevation at a time $t$ is then simply computed in the upstream direction, starting by the river outlets, as:

$$z(l,t) = z_R(l,t) + U_{mem}(l,t)\,(\tau(l,t) - \tau_R(l,t)) \text{ for } l > 0, \text{ and } z(0,t) = z_{base}, \tag{7}$$

As in the heterogeneous uplift case, this solution is easily implemented in a LEM by updating the river network and its properties after each time step. I also emphasize here that previous model (Fig. 4) is already a specific case of a time-variable uplift rate, with a change in uplift rate which occurs at the beginning of the simulation, leading in turn to a simpler formalism (Eq. 6). In the following, the paper is focused on demonstrating the ability of the model to simulate and track knickpoints. Indeed, discrete temporal changes in uplift rates or in base-level elevation can lead to sharp ruptures in the slope of river profiles, generally referred to as knickpoints (e.g., Rosenbloom & Anderson, 1994; Whipple & Tucker, 1999; Steer et al., 2019). Finite difference solutions to the stream power equation inherently lead to a progressive numerical diffusion of knickpoints during their migration, even with $n = 1$, while the algorithm developed here preserves the shape of knickpoints. To illustrate this advantage, I run a simulation with the same parameters than in the steady-state case, except that $U$ is raised from 10 mm.yr⁻¹ to 20 mm.yr⁻¹ at 250 kyr, for a total model duration of 500 kyr (Fig. 5). Compared to previous models, the model is here restricted to an extent of 10 over 2 km, with only one boundary (left) that is considered as possible outlets for

water. This setting limits fluvial network reorganization (or time-discontinuity) and in turn allow to track geomorphological features during the evolution of the landscape. I use a time step of 25 kyr, which is about 500 times greater than the time step imposed by the Courant condition, clearly above the range of time steps compatible with numerical solutions. Despite this, the knickpoints formed at the outlets of the model at 250 kyr, at the onset of the increase in uplift rate, are accurately modeled, i.e. with analytical accuracy, throughout their propagation (Fig. 5c). The shape of the knickpoint is also kept throughout its migration. I also highlight here, that due to the model setting with only one outlet boundary which limits river reorganization (and time discontinuity), erosion rates are smoother than in Figure 4.

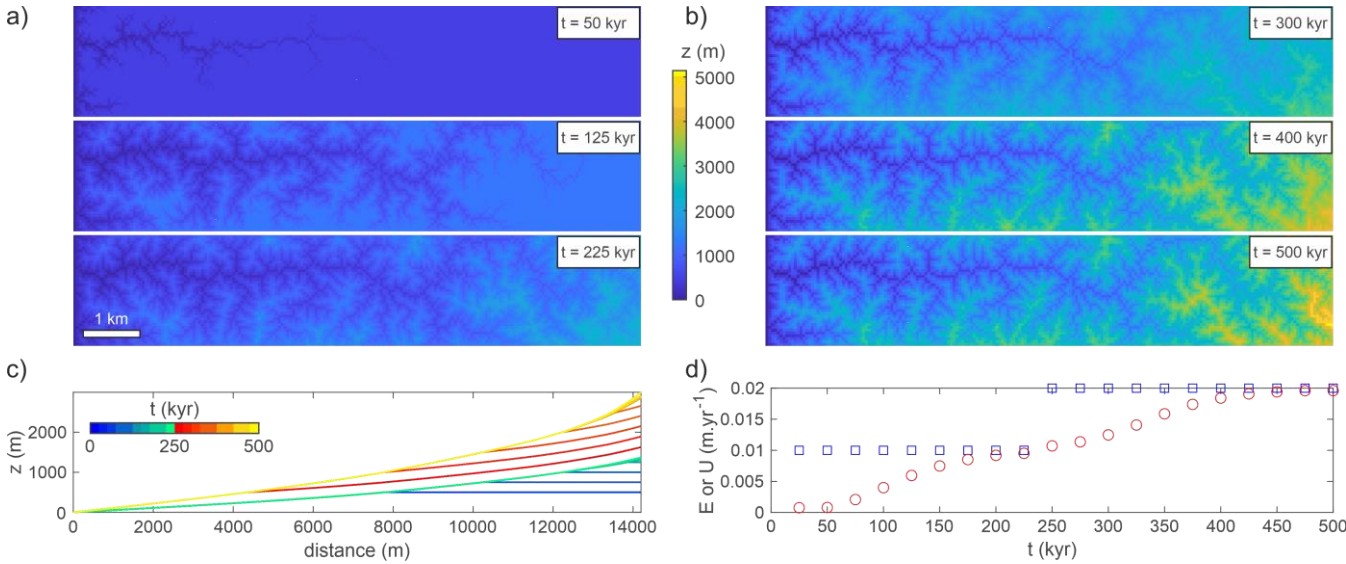

**Figure 5.** Dynamic evolution of the topography and knickpoint migration over 500 kyr. The initial uplift rate $U = 10$ mm.yr$^{-1}$ is doubled after 250 kyr. a) Topography before the increase in uplift at 50 (top), 125 (middle) and 225 kyr (bottom). b) Topography after the increase in uplift at 300 (top), 400 (middle) and 500 kyr (bottom). c) Temporal evolution of the longest river profile shown at every time step (except the first one), with the "winter" and "autumn" colormap showing river profiles before and after the increase in uplift. d) Temporal evolution of the uplift (blue squares) and erosion (red dots) rates.

**6 Solving for river and hillslope dynamics**

In previous sections, I have considered the steady-state and dynamic solutions of landscapes subjected only to river erosion following the SPIM. However, these analytical solutions can be extended to simulate the dynamics and morphology of colluvial valleys and hillslopes. Indeed, a power-law scaling for the slope-area relationship is observed in colluvial valleys, which suggest they could obey a similar erosion law than equation (1), but with a different couple of $m$ and $n$ exponents (Lague & Davy, 2003). A solution with $m = 0.24$ and $n = 1$, but considering a non-negligible incision threshold, was found to best explain the geometry of colluvial valleys in the Siwaliks Hills of Nepal for drainage area between $7 \times 10^{-3}$ and 1 km$^2$, representing the thresholds in drainage area between colluvial valleys and hillslopes or rivers, respectively (Lague & Davy, 2003). Below the area transition between colluvial valleys and hillslopes, the power-law scaling for the slope-area gets flat, due to landsliding and mass wasting processes, or reverts where hilltops are convex (Ijjasz-Vasquez and Bras, 1995; Tarolli &

Dalla Fontana, 2009). Once again, this hillslope domain could be geometrically modeled using the SPIM with a different couple of $m$ and $n$, for instance with $m = 0$ and $n = 1$ to model hillslopes following a critical angle of repose $S_c$. I do not argue here that these laws necessarily encapsulate the processes controlling colluvial and hillslope erosion (e.g. Tucker & Bras, 1998; Densmore et al., 1998; Roering et al., 1999; Lague & Davy, 2013; Jeandet et al., 2019), but that this framework can
5    approximate the observed geometrical relationships between slope and area.

Practically, considering three different erosion laws, for river, colluvial valleys and hillslopes, simply requires changing the value of $K$, $m$ and $n$ in the definition of celerity in the response time equation (Eq. 3) for each of the different domain, separated by thresholds in discharge or drainage area. Keeping $n = 1$ for simplicity leads to the following set of response time equations:

$$\tau(l) = \int_0^l \frac{1}{K_1(l)A(l)^{m_1}} \, dl' \text{ for } l < l_1 \tag{8}$$

$$\tau(l) = \int_0^{l_1} \frac{1}{K_1(l)A(l)^{m_1}} \, dl' + \int_{l_1}^l \frac{1}{K_2(l)A(l)^{m_2}} \, dl' \text{ for } l_1 < l \le l_2$$

$$\tau(l) = \int_0^{l_1} \frac{1}{K_1(l)A(l)^{m_1}} \, dl' + \int_{l_1}^{l_2} \frac{1}{K_2(l)A(l)^{m_2}} \, dl' + \int_{l_2}^l \frac{1}{K_3(l)A(l)^{m_3}} \, dl' \text{ for } l > l_2$$

where $A(l_1)$ and $A(l_2)$ are model parameters that define the threshold areas for river to colluvial valley and for colluvial valley to hillslope transitions, and $(K_1, m_1)$, $(K_2, m_2)$, and $(K_3, m_3)$ are the $K$ value and $m$ exponent for rivers, colluvial valleys and hillslopes, respectively. I emphasize here that the colluvial law used here is only inspired from the colluvial law describe in

15   Lague & Davy, 2003), as it neglects the incision threshold which lead to a non-linear behavior. Figure 6 shows the steady-state topographies obtained when considering river, colluvial and hillslope erosion. Considering these additional erosion laws therefore lead, as expected, to different scaling in the slope-discharge relationships, separated by thresholds in discharge or drainage area. These thresholds should be chosen to ensure 1) the continuity of the slope-discharge relationship and 2) the slope is equal to $S_c$ when $A \le A(l_2)$. I emphasize, once again, that the models developed here lead to slope-discharge

20   relationships with exact accuracy, at steady-state, due to the use of analytical solutions. Other analytical solutions can be considered to account for hillslope processes such as the one developed in the DAC model (Goren et al., 2014b).

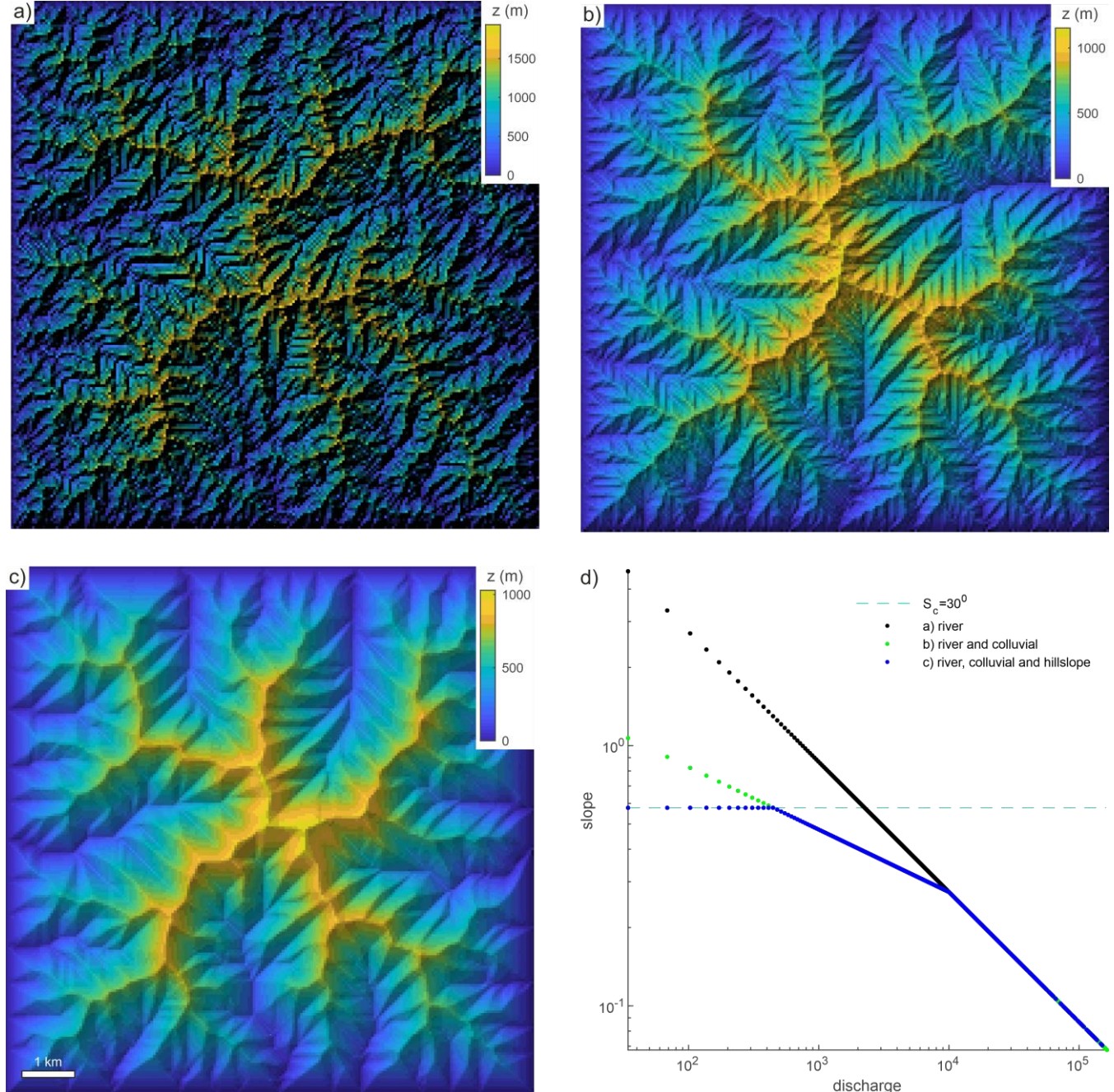

**Figure 6.** Steady-state topographies obtained with Salève considering only a) stream power incision ($m$ =0.5) in rivers (like in Fig. 1), b) stream power incision ($m$ =0.5) in rivers and colluvial erosion ($m$ =0.24) and c) stream power incision in rivers ($m$ =0.5), colluvial erosion ($m$ =0.24) and hillslope following a critical slope ($m$ =0) of $S_c$ =30°. To better highlight relief, elevation is represented by transparency over the raster of hillshade. d) Slope-discharge distributions for these three models.

# 7 Discussion and conclusion

Based on analytical developments (e.g. Royden & Perron, 2013), I have designed a new method to solve for the steady-state topography or the dynamic evolution of a landscape following the SPIM in 2D with analytical precision. The model can solve in a single time step, using an iterative scheme, the steady-state topography of a landscape under homogeneous or heterogenous conditions (i.e. uplift rate, erodibility and runoff). Iterations are required to optimize the planar organization of the river network and crest positions, starting from a random network. The number of iterations required for the convergence of the scheme only depends on the number of nodes discretizing the surface topography and only scales with $n_{pt}^{0.5}$, independently of other model parameters. Moreover, the model can also solve for the dynamic evolution of a landscape under either heterogeneous but constant or time-variable but homogeneous conditions. The dynamic and steady-state Salève models can solve for river, colluvial and hillslope erosion, if the associated erosion laws lead to slope-area (or discharge) relationships that can be modelled using a linear SPIM. The two main benefits of this new model are 1) its analytical accuracy that enables to suppress numerical diffusion and for instance to maintain the shape of knickpoints, and 2) the absence of an upper bound for the time-step that is not limited by the Courant condition. Contrary to any other state-of-the-art LEMs using the SPIM (e.g. Braun & Willett, 2013; Carretier et al., 2016; Campforts et al., 2017; Hobley et al., 2017; Salles, 2018), time stepping strategy in Salève can be chosen only based on physical considerations, such as the rate of river network reorganization, and not on numerical ones. All these advantages make Salève unique in its ability to efficiently model landscape evolution. In addition of its use in landscape evolution modelling, Salève could offer new opportunities to generate terrains for applications in computer graphics (e.g. Cordonnier et al., 2016), to infer the time and space evolution of uplift by inverting landscapes in 2D (e.g. Pritchard et al., 2009; Roberts & White, 2010; Goren et al., 2014a; Fox et al., 2014; Croissant & Braun, 2014) including river, colluvial valleys and hillslopes, to predict thermochronological ages from landscape evolution (e.g. Braun et al., 2014) or to validate the accuracy of numerical schemes used in other LEMs. The model is fast as it makes profit of the optimized flow routing algorithm provided by Topotoolbox (Schwanghart & Scherler, 2014).

The developed scheme, that uses 1D analytical solutions, is however limited to a flow networks that can be topologically classified as 1D node stacks or graphs (Braun & Willett, 2013), as resulting from a steepest slope flow routing algorithm. This excludes for instance recent models accounting for physical-based flow algorithms (Davy et al., 2017). The main limitation of this new approach is that reorganizations of the river network, such as catchment piracy, will not lead to transient phases of erosion, as the river elevation is directly updated to its optimal elevation for each node where $t \leq \tau(l, t)$. The response time of the landscapes is therefore shorter than with other 2D LEMs as it is equal to the 1D response time based on the flow network length at steady-state. Moreover, the flow network topology is updated at every iteration or time-step, in the steady-state or dynamic mode respectively, while other strategies, based on physical criterion, could be adopted (Goren et al., 2014b). If many dynamic LEMs use the same approach (e.g. Braun & Willett, 2013), this is a critical aspect of the convergence speed and computational time in the steady-state mode and future work should focus on accelerating it.

The Salève model is also not designed for horizontal tectonic displacement (e.g. Braun & Sambridge, 1997; Steer et al., 2011; Miller et al., 2007) that displaces nodes relatively to the location of the base-level condition. Moreover, Salève is a purely detachment-limited model which does not consider the role of sediment transport and deposition in landscape dynamics. Only the linear SPIM with $n = 1$ has been considered in this study, while some observations support non-linear models with greater values for $n$ (e.g. Lague, 2014). These limitations also emphasize that analytical solutions to landscape dynamics, such as Salève, represent a complementary approach to other "numerical" LEMS, which are by essence more versatile and allow to tackle coupled or complex scientific problems which characterized geomorphological systems.

Extending the Salève algorithm to non-linear SPIM represents a challenging and non-trivial perspective that requires to account for more complex analytical solutions with overlapping or stretching river profiles for $n > 1$ or $n < 1$, respectively (Royden & Perron, 2013). Using Salève to simulate the impact of both a heterogeneous and time-variable uplift rate has not been attempted and might also result in convergence issues. Moreover, using an even more efficient algorithm to route water also represents a promising avenue (e.g. Barnes et al., 2014). This is critical for Salève that can use a time step much greater than the Courant condition and which main computational limit is the flow routing algorithm. Therefore, no computational time benchmark was done for this new model, as the computation of elevation changes even on large grid is negligible compared to flow routing. In turn, solving for individual time steps in this model takes a similar amount of computational time than in other similar LEMs using the same flow routing algorithm (e.g. Braun & Willett, 2013; Schwanghart & Scherler, 2014). Yet, the advantage of this new model is its ability to use longer time steps while preserving analytical accuracy and consistency. Lastly, Salève represents the first attempt to use analytical solutions to model the dynamics of landscapes in 2D using the SPIM. Because little modifications are required to implement this solution in other LEMs, I believe the strategy developed in this paper could be adapted and further developed to make LEMs more efficient and accurate.

### Acknowledgments, Samples, and Data

Liran Goren and Sébastien Carretier are acknowledged for constructive reviews that helped improving this manuscript. I am also grateful to Sean Willett for his insightful comments on an earlier version of this manuscript. I thank Dimitri Lague, Philippe Davy, Jean Braun, Boris Gailleton, Joris Heyman, Alain Crave, Thomas Croissant and Edwin Baynes for their helpful comments and for discussions about this work. This project has received funding from the European Research Council (ERC) under the European Union's Horizon Horizon 2020 research and innovation programme (grant agreement No 803721). A Matlab version of the model (Steer, 2021) can be accessed through a GitHub and a Zenodo repository: https://github.com/philippesteer/Saleve_regular and https://zenodo.org/record/4686733. It is delivered with a routine to solve for the stream power law using an implicit finite difference solution.

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
