# Peer review of "Short communication: Analytical models for 2D landscape evolution"

_Earth Surface Dynamics, 2021_

## Author Response (AR1)

**Dear Editor,**

Your will find below my responses to the comments made by Liran Goren and Sébastien Carretier. I am very grateful for their positive comments. My responses are in blue. The changes made to the manuscript are first detailed after each response and then appear as comments on the manuscript with the track of changes.

Best regards,

**Philippe Steer**

**Review RC1 by Liran Goren**

Comment 1: The manuscript presents the development of a landscape evolution model (LEM) based on analytic solutions to the stream power incision model (SPIM), a simplified representation of vertical fluvial incision. Despite its simplicity, the SPIM is widely used in landscape evolution models for its demonstrated capabilities in reconstructing fluvial relief in response to changing boundary conditions at a relatively large spatial scale (i.e., Venditti et al. 2020). Model simplicity lent itself to explicit analytic solutions. Recently, there has been a growing interest in the behavior of the analytical solutions as part of studies that explored the expected dynamics of river long profiles as a function of changing geomorphic parameters and boundary conditions (i.e., Royden and Perron, 2013) and inversion applications (i.e., Goren et al. 2014a, Fox et al. 2014, Rudge et al. 2015, and many more).

The manuscript is well-written and clear. The motivation behind the research is well justified: developing an infinitely accurate 2D forward model for the evolution of fluvial relief in response to changing boundary conditions (in the current study, only changing uplift rate is demonstrated).

Response 1: I am very grateful to Liran Goren for her comments.

Comment 2: However, model implementation and its demonstrated usefulness remain somewhat immature and could benefit from additional exploration. Furthermore, the presentation of the model operation suffers from some biases. Here I specifically refer to (1) The physical meaning of the greedy algorithm that searches for acceptable topologies; (2) The application of "heterogeneous but constant uplift rate U(I)"; and (3) The claim that the model also solves for hillslope dynamics.

Response 2: I answer to these three main comments below.

Comment 3: For any given topology of a fluvial drainage network and associated boundary conditions, the analytic solutions of the SPIM predict the 2D relief of all the nodes in the model for all times. There is no need for numerical iterations. The donor-receiver relation is needed only to calculate the response time, tau(x), numerically (including for time-invariant and space variable erodibility and precipitation). Once the response time is known, the solution for each node can be derived independent of the other nodes (see, for example, solutions in eq. 6, Goren et al. 2014a, and eq. 13, Goren 2016, for the case

where K and P vary in time). Therefore, for any given topology, Saleve produces a graphical representation of the analytic solution.

**Response 3: I fully agree with this comment which does not call for any change in the manuscript.**

Comment 4: What the analytic solutions cannot predict is the drainage network topology. Here is where the model could become more significant. Saleve addresses it by updating the topology using a greedy algorithm that attempts to minimize chi (or potentially, chi\*, Willett et al. 2014, eq. 5) gradients across divides (in the manuscript, presented as elevation gradients), by updating the topology following the steepest descent in consecutive "update steps". The execution of each of these steps and the number of steps needed to achieve a stable topology have no physical time associated with them. While this issue is acknowledged in the manuscript, I'm not sure it is sufficiently discussed. Updating the topology toward the steepest descent commonly mimics drainage reorganization. However, whether each node should drain in the direction of its steepest descent neighbor regardless of the model spatial resolution is questionable. I.e., for sufficiently large grid spacing, this is probably not a very good representation. Additionally, there is no process related to this reorganization and, as stated in the manuscript, no time scale associated with it. Many LEMs use the steepest descent criterion to update the network topology (not all though, the DAC LEM (Goren et al., 2014b), for example, uses a physical criterion to decide if reorganization should take place or the older topology should be maintained) without discussing its meaning and related time scale. This omission is particularly apparent in Saleve because topological convergence based on the steepest descent algorithm while aspiring to minimize delta z across divides is the main numerical operation of the model.

Response 4: I partly agree with this comment. First, I agree that despite its popularity the steepest descent algorithm is not optimal to model flow over a topography. However, I do believe this algorithm is relatively incorrect at most scales, including small and large ones. We are currently, with some collaborators, making some efforts to develop better algorithms to account for flow hydrodynamics in LEMs following the initial work by Davy et al. (2017) and Croissant et al. (2017). However, many current LEMs still (and will continue to) use the steepest descent flow algorithm thanks to its simplicity. The scope of this paper is not to upgrade the flow algorithm of these LEMs, nor to tackle the issue of the consistency of the steepest descent flow algorithm with natural flows, but rather to demonstrate that the numerical solvers that most LEMs use for erosion can be changed to analytical solutions in some specific scenarios.

Liran Goren is perfectly correct when she mentions the absence of a timescale associated to the "update steps". However, this is not a major issue because:

1) in the steady-state mode, these "update steps" should not have any timescale associated to them (the model is not solving for a time evolution of a topography but for a static state at steady-state)

2) in the dynamic mode, the process of updating the flow network topology after each time-step is the classical approach that most LEMs follow (e.g. Braun & Willett, 2013), as also mentioned by Liran Goren.

I agree that other algorithms, such as the one used in DAC, could be used in future studies to decide whether to update the flow network topology.

Change to the Manuscript: I now discuss (in section 7 – Discussion and conclusion) in moredepth the potential issue of using the steepest descent flow algorithm and mention that morephysical algorithms could be used instead, as suggested by Liran Goren.

Comment 5: Moreover, an interesting case study, which is not attempted in the current manuscript (the author might want to consider including it), is when U varies in space and time. A convergence issue might arise in such cases: Each topology is associated with different tau(x) for each node x and a different max(tau(x)). Each topology, therefore, samples a different time range of the uplift rate history (from the present to max(tau) in the past), both generally, for the whole landscape, and specifically for each node, following eq. 7. Is there a way to guarantee topological convergence for such cases? Couldn't there be scenarios where the topology jumps between different configurations that sample different U histories without converging?

Response 5: If I agree this could represent an interesting additional test, the paper is already long for a short communication.

Change to the Manuscript: Therefore, I have added some a discussion sentence (in section 7) on this scenario to mention that the algorithm might have some convergence limits.

Comment 6: Section 4 starts with a declaration that the model input would be "potentially heterogeneous but constant uplift rate U(I)". In practice, in this section, the model is run with a uniform and constant U. The outcomes do not differ from those of section 5 with a time-variable uplift rate. The reason is that section 4 implicitly assumes that before U = 10 mm/yr, U was 0. This means that there is a temporal change (and no spatial change) in U, that generates a knickpoint, much like those in section 5. In fact, Figure 4c shows 2 knickpoints: One that corresponds to the transition from U = 0 to U = 10 mm/yr and the second from U = 10 to U = 20. The present form of section 4 is therefore redundant.

Response 6: I agree with this comment by Liran Goren that some redundancy exists between section 4 and 5. At the same time, the paper is also designed to add, step by step, some complexity. If it is correct that the model presented in section 4 already includes a temporal variation of uplift rate (as the model starts over a flat surface representative of U=0), section 4 is mostly designed to explain how the model, which is first used in a steady-state mode, can be used in a dynamic mode (which by essence is obtained by considering a change in uplift rate at the beginning of the simulation). Moreover, section 5 considers the general case of variations of uplift rate during the simulation itself which lead to a more complex solution (i.e. the "uplift memory map" approach) than the one presented in section 5. Section 5 is also focused on highlighting the quality of the model to track knickpoints in the Salève simulations, which is an application of the model developed in section 4.

Change to the Manuscript: I have now clarified the role of each section by mentioning that section 4 already exhibits a case of time-variable uplift rate, and that section 5 is an application dedicated to investigate the quality of the model to simulate and track knickpoints

Comment 7: Hillslopes are suggested to be represented by the SPIM with m=0, giving rise to a constant critical slope. To maintain a constant slope over the grid, with information propagation based on the response time, the erodibility of hillslopes, K\_3, should be infinite. This is a reasonable approximation, although a physically weird concept, for critical angle hillslopes, but there is no need for a numerical solver to represent it. These are just constant slope lines that adjust instantaneously to changes in elevation at their base (I\_2).

Response 7: I disagree with this comment.

First, K3 should not be infinite. Indeed, to guarantee that erosion rates are the same everywhere, the model must impose a continuity of erosion rates at the location A(I1) and A(I2), the transitions between hillslope, colluvial and fluvial domains. This imposes that:

E(l1)= K1 A(l1)^m1 S^n = K2 A(l1)^m2 S^n, and thus that K2=K1 A(l1)^(m1-m2)

E(l2)= K2 A(l2)^m2 S^n = K3 A(l2)^m3 S^n and thus that K3=K2 A(l2)^(m2-m3)=K2 A(l2)^m2 (as m3=0)

It gives, K3=K1 A(l1)^(m1-m2) A(l2)^m2, which is generally not infinite

Second, the benefits of using a response time for the hillslope domain, despite being a constant slope, is that it allows to manage all the model nodes using the same simple functions, which reduces model complexity. Moreover, it allows to consider potentially other formalism for hillslope erosion without requiring redeveloping a specific function. I agree that further developments could be made, such as using the interesting DAC approach for hillslopes (Goren et al., 2014), but I believe this is not mandatory for this manuscript that was submitted as a "Short Communication" paper.

Change to the Manuscript: I now acknowledge, as a perspective, the coupling of Saleve with DAC at the end of section 6.

Comment 8: To summarize, the concept of an infinitely accurate LEM, based on analytic solutions is appealing. However, the implementation presented in the current contribution raises some doubts regarding the model's usefulness for the landscape evolution community.

Response 8: I hope I have managed to convince Liran Goren of the usefulness of the developed approach. I thank here once again for her overall positive review.

**Line comments:**

Comment 9: Page 7, line 4. The concept of "thresholding the response time so that for every node  $\tau(l, t)$  = min( $\tau(l), t$ )." is not clear. If it meant to address the case when the response time is > max time for which there is information about the uplift, then this is like assuming U = 0 before information is available. This is a specific choice. Why assume U = 0 and not any other value?

Response 9: Indeed, this is equivalent to assuming that U=0 before the simulation starts. This is what most LEMs also do when they start with a flat initial surface. The objective here is to reproduce the settings of these models.

Change to the Manuscript: To clarify this point, I now explicitly mention in section 4 (just after equation 6) that "Thresholding the response time enforces that the uplift rate is considered null before the beginning of the simulation."

Comment 10: The discussion of the courant number (page 7, around line 20) is a bit over-emphasized. The analytic solution indeed has no inherent time limitations. As stated, it can be used to find the relief structure for any random time given the topology and the history of U.

Response 10: It is an important result of this new model to have no numerical limit on the time-step used.

**Review RC2 by Sébastien Carretier**

Comment 11: Philippe Steer presents a numerical solution scheme for the stream power incision model (SPIM) based on analytical solutions. This solution allows to reduce the computation time, and to preserve the shape of the knick-points, which are the two main advantages of this new model called Salève. The numerical solution proposed here does indeed offer perspectives for the inversion of topographies that would be demonstrably controlled by SPIM. Despite these interesting perspectives, I have several questions and remarks whose consideration could improve the clarity of this manuscript.

Response 11: I am very grateful to Sébastien Carretier for his comments.

Comment 12: The first one concerns the evolution of the drainage network. In the simplest scenario with constant uplift, it is indicated that the rivers develop according to the same network as in the initial stage (p5 line3). I confess that I do not understand this, neither in the text nor in figure 1 which shows that the network varies during the iterations. It is also obvious that the network must adapt dynamically since the initial topography is noisy and without any connected network. So I probably missed something here.

Response 12: I agree with Sébastien. The misunderstanding comes from the unclear explanation given in my manuscript. During the first iteration, the algorithm indeed uses the flow network of the initial iteration, as correctly pointed out by Sébastien Carretier (that is what line 3 of page 5 was meant to explain). However, the flow network is then updated during each iteration, so that the flow network at steady state is not similar to the initial flow network.

Change to the Manuscript: I have clarified the explanation about the flow network in section 3 and added a figure (Fig. 2a) to explain the main stages of the algorithm.

Comment 13: I have the same question about the transient simulations where it is said that the horizontal reorganization of the network is instantly accompanied by a topographic adaptation. How and why does this reorganization of the drainage network in Salève take place? I think that further

explanation of the procedure by which the drainage network is established at a given iteration, or a given time, is strongly required.

Response 13: Indeed, compared to classical LEMs which compute erosion to update topography, Saleve directly computes a topographic state at a time t based on the current flow network. In turn, any change to the flow network, between two time-steps, will directly lead to an instantaneous change of the topography. This is already largely mentioned in the manuscript (in section 4).

The flow network is computed as in most LEMs based on the topography at the previous time step, which defines the steepest slope and the flow network. The flow network can change significantly in between successive iterations due to river capture.

Change to the Manuscript: I believe the existing explanations as well as the new figure (Fig. 2b) clarify this point.

Comment 14: Concerning the scope of this numerical solution, it is well specified that it is limited by the value of n=1, which could be different in natural cases. It would also be good to add that this model is a pure erosional model, without deposition and therefore a pure detachment limited model. The absence of sedimentation, which controls the degree of limitation by transport or detachment (Davy and Lague, 2009), could be discussed as one of the strong limitations.

Response 14: I agree, even if the detachment-limited nature of this model was already mentioned in section 2.

Change to the Manuscript: I have therefore clarified these limitations in the Discussion and conclusion section. "Moreover, Salève is a purely detachment-limited model which does not consider the role of sediment transport and deposition in landscape dynamics. Only the linear SPIM with n=1 has been considered in this study, while some observations support non-linear models with greater values for n (e.g. Lague, 2014)."

Comment 15: The treatment of colluvium erosion is treated here using a different value of m, referring to Lague and Davy (2003), but in that paper the erosion law includes a large erosion threshold, which results in an n>1 in the SPIM, whereas the law used in the Salève uses n=1. It is therefore questionable to argue that slope erosion is taken into account by changing only m.

Response 15: I agree with this comment. Indeed, I took a "shortcut" by neglecting the threshold in Lague & Davy (2003) colluvial erosion law. Using their parametrization, this threshold is of prime importance for low to intermediate value of uplift rates (up to a few cm/yr) and become less and less important for greater uplift rates.

Change to the Manuscript: As the objective is to illustrate that the model can be extended to domain other than the fluvial one, I have decided to keep this part. I however now clearly mention in section 6 that the model I use is inspired by the colluvial law of Lague & Davy (2003) but does not integrate the threshold effect.

**Specific comments:**

Comment 16: Page 7 line 7. Could you explain what do you mean by "non-optimality of the planar organization of the river network" ?

Response 16: I mean that the flow network is not already in its steady-configuration starting from the first time-steps. This is important as once again, the only limitation in terms of time step for this analytical model, is the need to update the flow network "frequently". Otherwise, the topography can exhibit sharp height differences at the crest between successive catchments.

Comment 17: Page 7 line 18 "than" -> as?

Response 17: Done

**Comment 18: Page 9 line 27. Could you explain what do you mean by "slope patches"?**

Response 18: Slope patches is a term that was coined by Royden & Perron (2013): "A river longitudinal profile evolving according to the stream power law consists of a series of contiguous segments, or "patches," of slope." (section 5). We already refer to this explanation in section 2 of the paper.

**Comment 19: Page 11 I suppose you specify 11, 12 and 13?**

Response 19: Yes, l1 and l2 are parameters that need to be specified. It is now mentioned that these are model parameters.

Comment 20: Page 13 line 14 Carretier et al. 2016 (not 2015)-> the same in the biblio list

Response 20: Done